# High-quality sugar production by *osgcs1* rice

Yujiro Honma[1,2,3,8], Prakash Babu Adhikari [1,2,8], Keiko Kuwata[4], Tomoko Kagenishi[3], Ken Yokawa[3], Michitaka Notaguchi [4,5], Kenichi Kurotani[5], Erika Toda [6], Kanako Bessho-Uehara [5,7], Xiaoyan Liu[1,2], Shaowei Zhu[1,2], Xiaoyan Wu[1,2] & Ryushiro D. Kasahara [1,2,8 ✉]

Carbohydrates (sugars) are an essential energy-source for all life forms. They take a significant share of our daily consumption and are used for biofuel production as well. However, sugarcane and sugar beet are the only two crop plants which are used to produce sugar in significant amounts. Here, we have discovered and fine-tuned a phenomenon in rice which leads them to produce sugary-grain. We knocked-out *GCS1* genes in rice by using CRISPR technology, which led to fertilization failure and pollen tube-dependent ovule enlargement morphology (POEM) phenomenon. Apparently, the *POEMed-like* rice ovule ('endosperm-focused') can grow near-normal seed-size unlike earlier observations in *Arabidopsis* in which *gcs1* ovules ('embryo-focused') were aborted quite early. The *POEMed-like* rice ovules contained 10–20% sugar, with extremely high sucrose content (98%). Trancriptomic analysis revealed that the *osgcs1* ovules had downregulation of starch biosynthetic genes, which would otherwise have converted sucrose to starch. Overall, this study shows that pollen tube content release is sufficient to trigger sucrose unloading at rice ovules. However, successful fertilization is indispensable to trigger sucrose-starch conversion. These findings are expected to pave the way for developing novel sugar producing crops suited for diverse climatic regions.

[1] School of Life Sciences, Fujian Agriculture and Forestry University, 350002 Fuzhou, Fujian, China. [2] FAFU-UCR Joint Center and Fujian Provincial Key Laboratory of Haixia Applied Plant Systems Biology, Haixia Institute of Science and Technology, Fujian Agriculture and Forestry University, 350002 Fuzhou, Fujian, China. [3] Faculty of Engineering, Kitami Institute of Technology, Hokkaido 090-8507, Japan. [4] Institute of Transformative Bio-Molecules, Nagoya University, Nagoya 464-8601, Japan. [5] Bioscience and Biotechnology Center, Nagoya University, Furo-cho, Chikusa, Nagoya, Aichi 464-8601, Japan. [6] Plant Development and Physiology Laboratory, Department of Biological Sciences, Tokyo Metropolitan University, Hachioji, Tokyo 192-0392, Japan. [7] Department of Plant Biology, Carnegie Institution for Science, 260 Panama street, Stanford, CA 94305, USA. [8]These authors contributed equally: Yujiro Honma, Prakash Babu Adhikari, Ryushiro D. Kasahara. ✉email: kasahara@fafu.edu.cn

We do not have too many choices among sugar producing crops other than sugarcane and sugar beet, which can produce significant amounts of efficiently extractable sugars[1–3]. Sucrose purity is crucial for its crystallization[4]. The sugar produced by other plants such as maple trees and sugar palm have significantly low sucrose content in their sap[5,6] and are not much efficient to produce table sugar. An alternative approach for producing sugary syrup is saccharification, which involves an ex vitro digestion of starchy grains and cellulose fibers to produce simple sugars such as glucose, maltose, and sucrose[7,8]. However, saccharification is still more labor intensive and less efficient as compared to the direct sucrose extraction from sugarcane or sugar beet. These sucrose-rich plants are also important sources of bioethanol production for which the quantity and quality of their sucrose content are crucial[9]. While both sugarcane and sugar beet fit into that parameter, niche of their cultivation is restricted to just either warm or cold region, respectively[10]. Thus, establishing a plant with broader niche of cultivation as a source of high quality (and quantity) of sucrose production using a process that avoids the need for saccharification would benefit both sugar and bioethanol productions. In this study, we discovered a novel sugar-producing rice phenotype, sugar rice, which contains high-quality sucrose in its ovule by halting its conversion to starch. Our findings offer an innovative approach for the sugar production in planta.

## Results and discussion

In previous study, we identified an important phenomenon, pollen tube-dependent ovule enlargement morphology (POEM), a new reproductive step between pollen tube guidance and fertilization[6]. Pollen tube content (PTC) release inside the ovule triggers POEM, which in turn, increases the ovule size and initiates seed coat formation without fertilization[11,12]. Discovery of the POEM phenomenon showed its potential applications in crop breeding for seed size increment and apomixis induction. However, POEM has, to date, only been reported in *Arabidopsis*. To investigate if this phenomenon is conserved in monocot as well, we took rice (*Oryza sativa*, *japonica* ssp. "Nipponbare") as a plant of interest in current study. Since *Arabidopsis gcs1* mutant showed clear POEM phenomenon in earlier study[11], we first conducted a homology search to see whether *O. sativa* possessed *GCS1*[13] homologs or not. *GCS1* stands for generative cell specific 1 and refers to a gene that is expressed specifically in the sperm cells. It is required for double fertilization between sperm cells and the egg/central cells in *Arabidopsis*. Two candidate homologs of rice *GCS1* were identified, those being *OsGCS1* (Os05g0269500) and *OsGCS1-like* (Os09g0525700) (Fig. 1a and Supplementary Fig. 1). *OsGCS1* has a signal sequence (SS) at its N-terminus and a transmembrane domain (TD) near its C-terminus, while *OsGCS1-like* lacks the SS and the TD is nearly absent as well. However, both have a relatively conserved HAP2/GCS1 domain (Supplementary Fig. 2). We developed genome-edited rice plants by knocking-out these two genes (Fig. 1a and Supplementary Fig. 2) using the CRISPR/Cas9 technology[14–16]. We additionally obtained a Tos17 mutant line for *OsGCS1-like* and observed its phenotype. Since *AtGCS1* is known to be expressed at significantly high levels in the sperm cells, a similar subcellular expression of *OsGCS1* and *OsGCS1-like* is expected in rice, as both genes have been revealed to be sperm cell-specific (Fig. 1b). A previous report[17] showed the specific expression of *GCS1* (Os09g0525700) in the sperm cells of a Japonica rice variety, which completely matches our observations of the *OsGCS1* expression pattern. Pollinated rice styles were stained with aniline blue, which showed that *osgcs1* pollen tubes have germinated and grown along the style until reaching the female

gametophyte, in a similar way to what happen with the Nipponbare pollen tubes. These results indicate that *osgcs1* mutants have no pollen tube guidance defects (Fig. 1c, d), and that possibly these pollen tubes burst to release the PTC, as reported in *Arabidopsis*. Next, we observed dissected grains of Nipponbare (Fig. 1e, f), *osgcs1* (Fig. 1g, h), and *osgcs1-like* (Fig. 1i, j) plants. Nipponbare seeds possessed an embryo and endosperm inside the aleurone layer. However, *osgcs1* and *osgcs1-like* seeds had neither embryo nor endosperm, suggesting that the enlarged seed-like tissue might not have been fertilized. We conducted the reciprocal crosses using *osgcs1* (*g51*) mutants and obtained 100% ($n = 6$ ovaries) fertilized seeds when the *osgcs1* (*g51*) mutant was pollinated with Nipponbare pollen and 100% ($n = 8$ ovaries) seed-like tissue (as shown in Fig. 1g–j) when the Nipponbare ovaries were crossed with *osgcs1* (*g51*) pollen, indicating that the mutation is transmitted from *osgcs1* male gametophytes. Based on these results, we concluded that the *osgcs1* and *osgcs1-like* mutations were independently responsible for rice POEMed-like phenotype. Interestingly, the seed-like tissues of both *osgcs1* and *osgcs1-like* mutants (Fig. 1g–j) were almost as large as that of Nipponbare seeds (Fig. 1e, f). However, unlike Nipponbare seeds, the mutants derived seed-like tissues contained transparent liquid (Fig. 1g, h, j). Based on our observations, the watery seed-like tissue should contain no starch since the liquid was transparent. The contents of the tissue are discussed below. The ratios of the watery seed-like tissue development are shown in Fig. 1k (*osgcs1*: 46–98%; *g28*: $n = 7$ plants, *g83*: $n = 5$, and *g51*: $n = 6$; *osgcs1-like* 42–44%; *g41*: $n = 5$ and Tos17_NC0320: $n = 4$). Due to stronger phenotype of *osgcs1* (*g51*) as compared to any other mutants, we took it for further observation in this study.

Since the *osgcs1* and *osgcs1-like* mutants showed the POEMed-like phenotype, as previously shown in POEMed *Arabidopsis*[11], we hypothesized that there could be both shared and species-specific processes involved in the POEMed and/or POEMed-like phenomena in these two species. To test this hypothesis, RNA-seq technology was used to analyze the transcriptomes of ovules from Nipponbare and *osgcs1* (line *g51*) plants using an Illumina sequencer. The data acquired were further explored in order to better understand the differences in the gene regulatory networks involved in starch and sucrose metabolism (Fig. 2a–e). We first performed cluster analysis to obtain an overview of the situation for the two. Hierarchical clustering indicated transcriptional similarity of Nipponbare and *osgcs1* ovules at 1 day after pollination (DAP) and 3DAP (Fig. 2a). Soon after pollination, however, 374 transcripts were up-regulated in both Nipponbare and *osgcs1* ovules indicating that PTC release itself is sufficient to trigger transcriptional changes in rice ovules (Fig. 2b, c and Supplementary Data 1, 2). As reported in *Arabidopsis*[11], multiple genes associated with cell expansion or cell division were significantly up-regulated in both Nipponbare and *osgcs1* ovules at 1DAP or later (Fig. 2d and Supplementary Data 3). Since POEMed-like rice ovules showed no starch phenotype (Fig. 1g, h, and Supplementary Fig. 3), we checked the expression of genes related to starch synthesis. Some of these genes revealed a relatively lower expression in *osgcs1* ovules compared to their Nipponbare counterparts (Fig. 2d). To further clarify the molecular event, we investigated gene expression patterns related to starch and sugar metabolism using the KEGG PATHWAY database (https://www.genome.jp/kegg/pathway.html). Out of 169 genes related to starch metabolism, our transcriptome analysis identified the expression on only 94 genes. Among these, 15 genes whose expression was increased in Nipponbare ovules after pollination, were not strongly expressed in *osgcs1* ovules (Supplementary Data 4, 5). These genes encode catalytic enzymes involved in sucrose hydrolysis or starch synthesis (Fig. 2e and Supplementary Fig. 3). These data suggested that PTC release

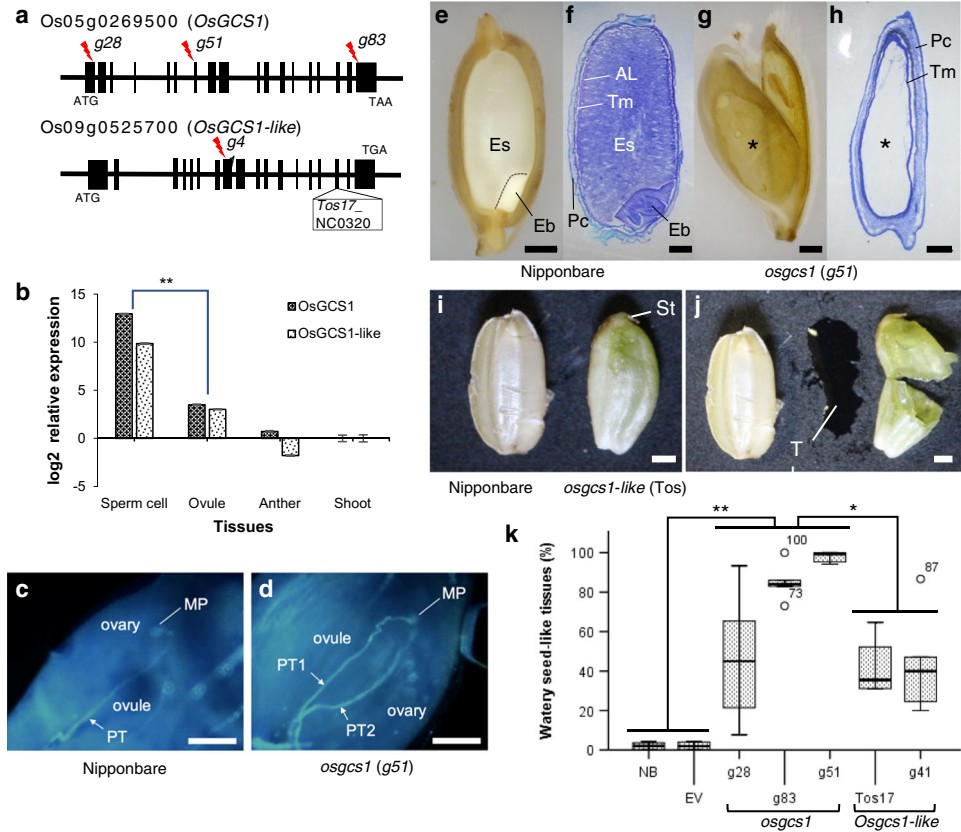

**Fig. 1 OsGCS1 and OsGCS1-like genes and phenotypes of their respective mutants. a** *OsGCS1* (Os05g0269500) and *OsGCS1-like* genes (Os09g0525700) with the mutation sites (*g28*, *g83*, *g51*, and *g41* with red lightning). Vertical bars (exons) and horizontal lines (introns). **b** Tissue-specific qRT-PCR expression analysis of *OsGCS1* and *OsGCS1-like* genes. *OsACT1* (Os03g50885) was used as an internal control. The bar whiskers represent SEM of two biological and six technical replicates (**$p < 0.001$ in two-tailed *t*-test). **c, d** Aniline blue staining. Only one pollen tube (PT) can be seen which is inserted to micropyle (MP) in Nipponbare ovary (1DAP) ($n = 24$ pistils) (**c**). In *osgcs1* (*g51*) mutant (1DAP), PT1 has inserted to micropyle, however, PT2 has not yet ($n = 68$) (**d**). *osgcs1* PTs had no PT guidance defects as in *Arabidopsis*[30]. Bars: 100 µm. **e, f** A Nipponbare grain. A border of the embryo (Eb) and endosperm (Es) (Dashed line). Pc pericarp, Tm tegmen, and AL aleurone layer. **g, h** *osgcs1* grain lacks embryo, endosperm and aleurone layer but transparent liquid (*). **f, h** Toluidine blue staining. Bars: 1 mm. **i** Nipponbare seed and an *osgcs1*-like (Tos) mutant seed-like structure with reminiscent of a stigma (St) ($n = 10$). **j** Transparent liquid (TL) from the structure. **k** Watery seed-like tissue production by *OsGCS1* or *OsGCS1-like* gene mutations. Each box represents the data at 25th percentile (lower end), median (horizontal line within the box), and the data at 75th percentile (upper end). The outer ends of whiskers at each box represent the maximum and minimum data points. Outliers, when present, are represented by small circle with associated data value (*$p \le 0.5$, **$p \le 0.05$, Tukey-Kramer multiple comparison test; $n = 10$ (NB Nipponbare), 10 (EV empty vector), 5 (*g28*), 5 (*g83*), 6 (*g51*), 4 (Tos), and 5 (*g41*)).

affects starch and sucrose metabolism gene-network in rice ovule before/without fertilization.

Judging from the transcriptome data, we speculated that the major component of the watery content of the POEMed-like *osgcs1*-ovule was unlikely to be starch. We analyzed the watery content in the ovules, focusing on saccharides. Interestingly, it showed 97.1% ± 1.0% sucrose, 1.8% ± 0.6% glucose, and 1.1% ± 0.3% fructose (mean ± SD, $n = 4$) (Fig. 2f). Of the total volume, the sugar rice contained 16.4% ± 6.9% sucrose ($n = 4$) (Fig. 2g). It strongly indicates that while released PTC enhances if not triggers the genes involved in sucrose synthesis and metabolism, fertilization is essential for its further conversion to starch in rice ovules, Furthermore, it shows that *osgcs1*-ovules are essentially "sugar rice" in which sucrose-starch conversion has been hindered. Figure 3 illustrates the proposed pathway by which sugar rice is produced.

Normally developing rice grain shows steep increase in sucrose accumulation shortly followed and dominated by starch deposition[18,19]. In addition, it has gradually increased activity of most of the enzymes involved in sucrose metabolism and starch biosynthesis[19]. In the current study, we observed the upregulation of gene involved in sucrose and UDP-glucose inter-conversion but downregulation of the potential genes involved in UDP-glucose to ADP-glucose and its further conversion to starch in *osgcs1*-ovules at 3DAP (Supplementary Fig. 3). Since, ADP-glucose is crucial for starch biosynthesis, its hindered production in *gcs1* ovule might have been affecting starch biosynthesis in the mutant ovules. Additional studies on the genes may confirm as such in the future. Analyses on some additional interesting genes have been provided in Supplementary Discussion.

The sucrose content we observed in the pseudograins of the sugar rice was comparable with that reported for sugarcane (98% sucrose, 1.0% glucose, and 1.0% fructose)[1] and sugar beet molasses (98% sucrose, and 2.0% glucose/fructose mixture)[2] indicating its potential usefulness in commercial sugar and/or biofuel production. The ever-increasing global sugars consumption at present is almost 180 million metric tons per year[20] and fuel consumption is above 100 million barrels per day. The continued if not increased supply of sugar-rich crop is important to support human consumption as well as for the biofuels and bioplastics production[21]. Sugarcane and sugar beet are the only major crops used in efficient sugar production[3], while the former

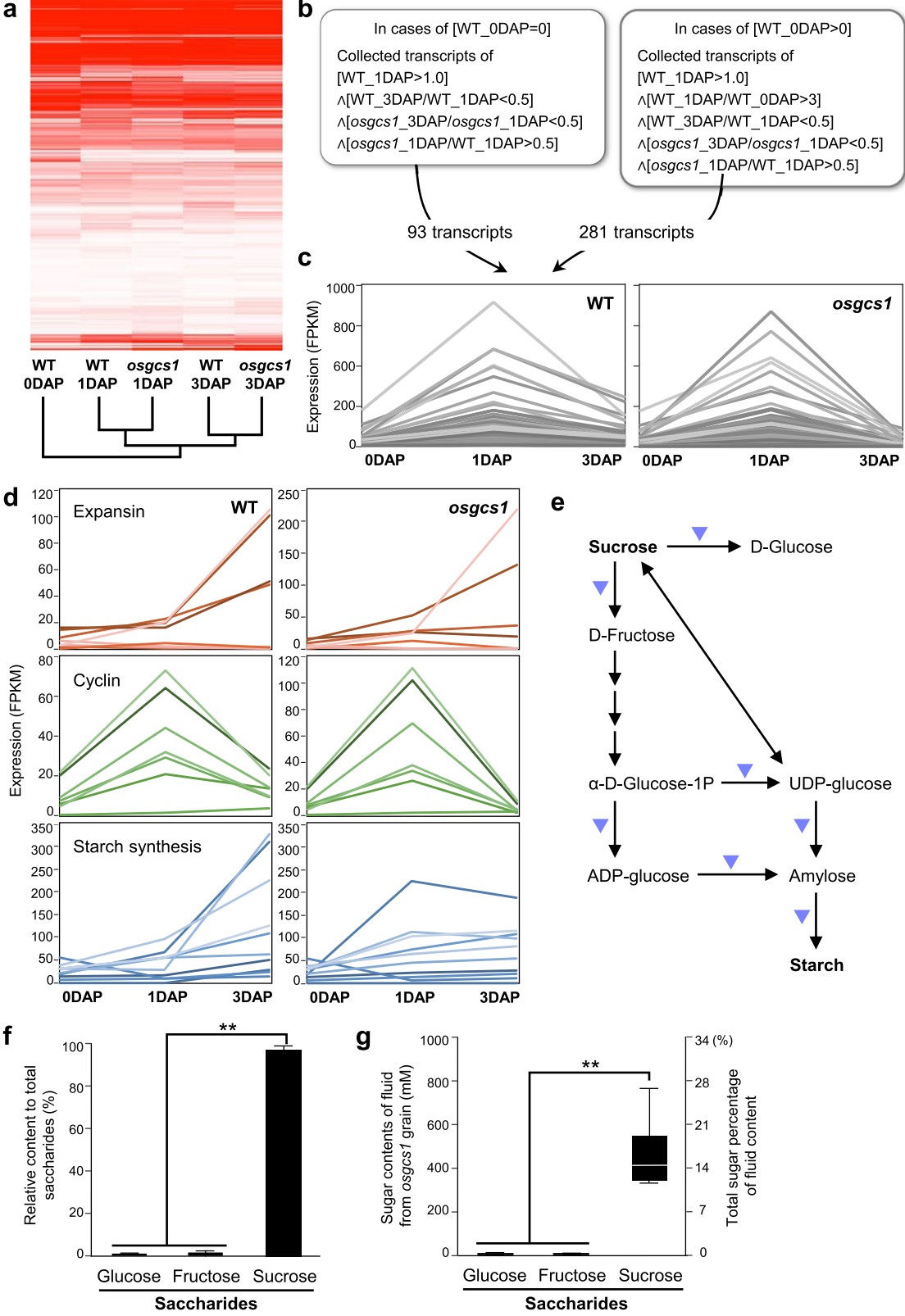

and maize are those major crops used in biofuel (ethanol) production with high efficiency[22]. Other plants used in the processes are not much efficient as these crops. Unfortunately, these crops can not be grown in broad climatic regions as maize and sugarcane is suitable for warm regions while sugar beet for cold region. Moreover, due to recent global extremes in weather conditions resulting in sudden drought, or abnormally hot or cold

temperatures, cultivation regions of various crops are getting seriously affected[23].

While there is an approach of reducing the green-house gas production and, at least, curbing the speed of climate change, another wise approach would be to brace ourselves for the worst. Dependency on narrow resources is an existential threat to any species. Having access to a wider range of sugar-producing crops

**Fig. 2 Typical gene expression pattern in the *osgcs1* mutant with high sucrose accumulation in its ovules. a** Hierarchical clustering of transcriptome data obtained from Nipponbare (NB) ovules before pollination (NB 0DAP), NB ovules at 1 and 3 days after pollination (NB 1DAP and NP 3DAP), and *osgcs1* ovules at 1 and 3 days after pollination (*osgcs1* 1DAP and *osgcs1* 3DAP). **b** Early-response genes triggered by PTC release were screened in the two classes indicated and pooled for further expression analyses in **c** (see "Method" for details). **c** Temporal expression of the identified genes in NB and *osgcs1* ovules. **d** Temporal expression of the genes for cell expansion (expansin), cell division (cyclin), and starch synthesis in NB and *osgcs1* ovules. **e** Metabolic cascade for starch synthesis from sucrose in rice. The expression of genes encoding the enzymes generating glucose or fructose from sucrose and the enzymes acting at steps further downstream for starch production were down-regulated in *osgcs1* ovules compared to NB ovules (marked in blue triangles). **f** Sugar component promotion of the fluid from *osgcs1* ovules. The fluid contained 1% glucose, 1% fructose, and 98% sucrose. The bar whiskers represent SEM of respective lines (**$p \leq 0.05$, Tukey's test; $n = 5$). **g** Sugar content of the fluid obtained from *osgcs1* ovules. Each box represents the data at 25th percentile (lower end), median (horizontal line within the box), and the data at 75th percentile (upper end). The outer ends of whiskers at each box represent the maximum and minimum data points (**$p \leq 0.05$, Tukey's test; $n = 5$).

plants would certainly benefit mankind at present as well as in the recent future. The technology of producing sugary-grain in the plants which would otherwise produce starchy seeds, we developed in the current study, can be applied in crops of diverse ecological niche thereby broadening our sugar-producing crop-resources. If the process necessary to harvest sugar rice were developed, it could contribute in global sugar production and we may not have to rely on only two crops (sugarcane and sugar beet).

In addition to its direct agricultural and commercial significance, the current study has offered a key information in sucrose metabolism and starch biosynthesis. Multiple enzymes involved in the process have already been identified (Supplementary Fig. 3) and we know that starch synthesis in rice (and other starchy crops') grain is followed by a short but sharp increase in sucrose accumulation after fertilization event[18,19]. However, it was yet unclear whether those two biochemical steps are triggered by independent or a single physical event. Apparently, these events are not as blurry as they appear. Our study strongly indicates that PTC release and nuclear fusion, the two successive but normally quick events of double fertilization process, are the respective cues to trigger phloem unloading of sucrose at the ovule and its further conversion to starch. Some mutants with dysfunctional starch biosynthesis have been identified in cereal crops which include R-type debranching enzyme (DBE) (*sugary-1, su1*), *branching enzymeI* (*beI*)*, beIIb,* and *amylose-extender* (*ae*) mutants of rice; *su1*, pullulanase-type DBE (*zpu1*), *starch synthase* (*ss*), and *waxy* (*wx*) mutants of maize; isomamylase-type DBE mutant (*isa1*) of barley etc[24]. However, unlike POEMed-like sugar rice, none of these mutants are fertilization defective as sucrose is metabolized to intermediary products but it can not be further converted into starch in their developing grains[23].

Additionally, the pericarp and tegmen (seed-coat equivalent in rice) of the near-normal-sized POEMed-like rice ovule were relatively thicker than those in its Nippombare counterparts at 10 DAP (Fig. 1e–i). Gradual thinning of these layers and their degeneration at the final stage of grain development is normal for Nippombare grains[25]. Their persisting structures at the POEMed-like rice ovules show that instead of thinning, these layers expand to fulfill the space requirement as the phloem unloading of sucrose continues in the ovule. Few upregulated expansin-related genes identified *via* transcriptome analysis (Fig. 2d and Supplementary Data 3) are the most likely candidates directly involved in the process. In earlier studies on *Arabidopsis*, Kasahara, et al.[11] and Liu, et al.[12], observed that PTC release alone can trigger the seed coat initiation but can not sustain ovular growth beyond 3DAP. The fundamental differences in the developmental process and function of the component tissues in rice and *Arabidopsis* seeds might be the underlying cause behind the disparity between their respective POEMed and POEMed-like ovules. In rice (a monocot), its caryopsis-type "endosperm focused" fruit/seed uses

endosperm as the energy source during the germination process. In *Arabidopsis* (a dicot), on the other hand, endosperm is reduced to a single-celled layer in its "embryo-focused" seed and seed-germination is supported by the energy conserved in the embryonic leaves (cotyledons). Apparently, the phloem unloading of sucrose at embryo-focused and endosperm-focused ovules are maintained differently. It is likely that the POEM phenomenon sustains the growth of endosperm-focused ovules for relatively longer duration than their embryo-focused counterparts. Additional comparative studies in the future will shed more light on the subject.

We are still far from explaining every molecular and physiological event, which occurs in developing seeds. The present study provides an insight in coordinated physical and molecular events during early steps of fertilization process in endosperm-focused rice seeds. The high sugar content observed in the POEMed-like rice ovules has immediate as well as far-reaching significance. The technology may potentially open a whole new avenue of developing broad range of high-quality sugar-producing crops (of different species) suitable for diverse ecological zones thereby broadening our sugar-crop resource base.

## Methods

**Plant materials and growth conditions**. The rice variety "Nipponbare" (*Oryza sativa* ssp. *japonica*) was used as the wild type (WT) for comparing the seed phenotype and as the source of the CRISPR/Cas9 lines. Tos17_NC0320, a T-DNA inserted line whose background is Nipponbare, was provided by the National Agriculture and Food Research Organization in Japan. Both lines were grown in Cera Fudou (Phytoculture Control Co, Ltd. Japan, http://www.phytoculture.co.jp/index.html), in a rice isolated growth chamber under 13 h light/11 h dark conditions at 28 °C day/20 °C night until flowering.

**CRISPR and transformation experiments**. Four pairs of primers were designed for respective sets of 20-nt guide RNA (gRNA) (three for *OsGCS1* and one for *OsGCS1-like*) (Supplementary Table 1, no. 1–8). The primers of respective gRNA were annealed and cloned into pU6gRNA-oligo vector after digestion with BbsI. Next, the gRNA cassette in the pU6gRNA-oligo vector was transferred into a gRNA/Cas9-expressing binary vector (pZH_OsU6gRNA_MMCas9)[14–16] at AscI and PacI restriction sites. The constructed binary vectors along with pZH_OsU6g-RNA_MMCas9 and empty vectors (as controls) were introduced into Nipponbare via *Agrobacterium tumefaciens* (strain: EHA105)-mediated transformation.

**RNA extraction from various tissues**. Tissue-specific total RNAs were extracted for qRT-PCR analysis as described by Abiko et al.[26] and Rahman et al.[27]. For the transcriptome analysis, total RNA was extracted from ovary samples surgically isolated from flowers without stigma 1 day and 3 days after flowering, with a single pistil collected in the mutant and 5 pistils in Nipponbare plants. The samples were ground with a pestle in a 1.5-ml tube on liquid nitrogen and total RNA extraction and DNase treatment were performed using RNAqueous-Micro Total RNA Isolation Kit (Thermo) according to the manufacturer's instructions. The RNA was quantified using Qubit™ 3 fluorometer (Thermo) and Qubit™ RNA HS Assay Kit.

**cDNA synthesis from various tissues for qPCR experiment**. cDNA for the sperm cells was synthesized and amplified as described previously[27]. For qPCR analysis, 0.5 μl of cDNA was used with LightCycler 480 SYBR Green I Master (Roche Applied Science, Penzberg, Germany) according to the manufacturer's

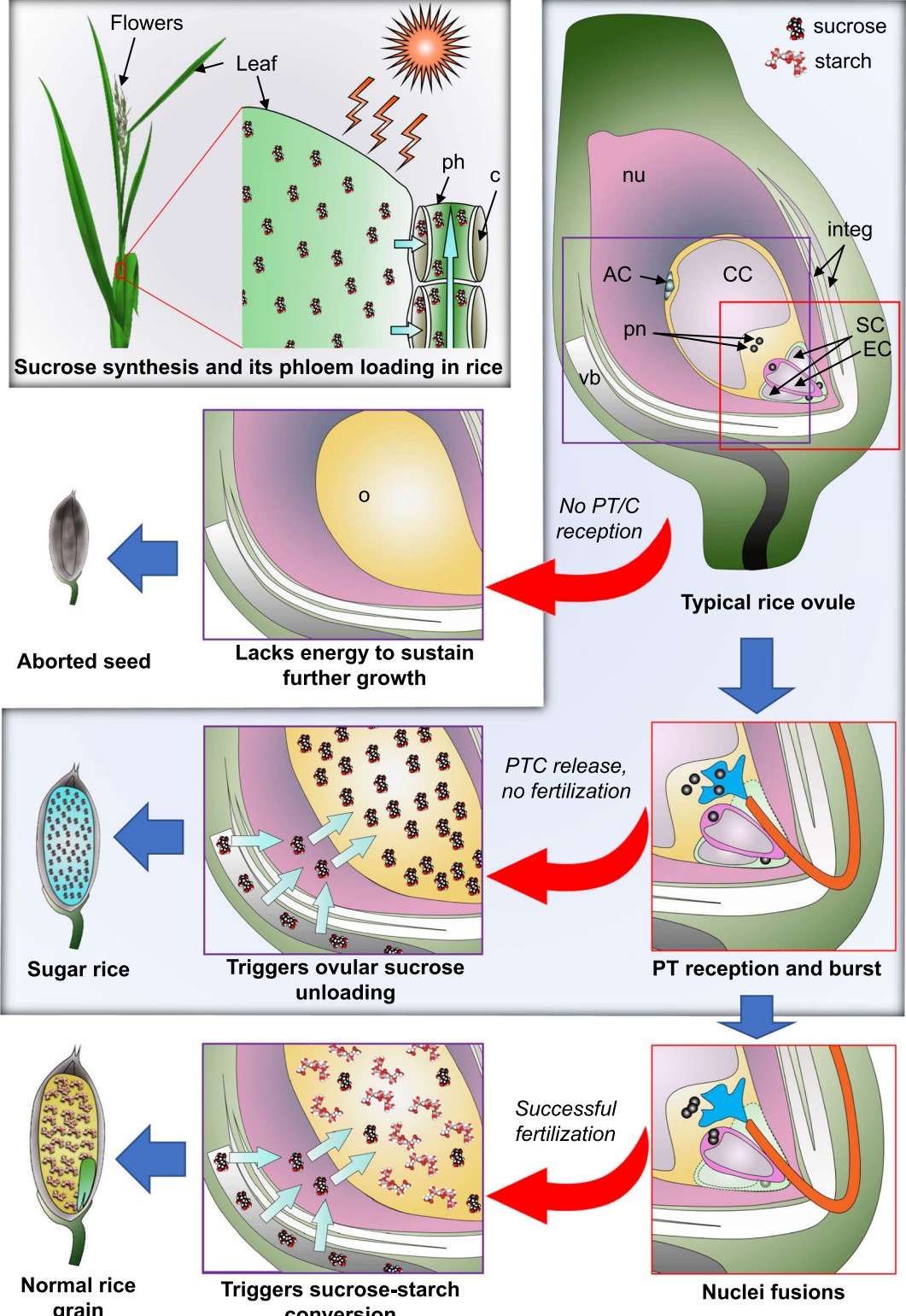

**Fig. 3 Proposed mechanism of sugar rice production.** Sucrose synthesized in rice leaves after photosynthesis is loaded to phloem. Its unloading into ovules should be triggered by pollen tube content (PTC) release. While successful fertilization after PTC release further triggers sucrose–starch conversion and development of normal starchy rice grains, the fertilization defect in *osgcs1* ovules should lead to the continued accumulation of sucrose and development of sugar rice. In absence of pollen tube or PTC reception, ovules cannot sustain their prolonged growth and are subsequently aborted. The purple square in the typical rice ovule is expanded to show sucrose unloading and/or sucrose–starch conversion in subsequent steps, while the red square has been expanded to show fertilization steps. The path of sugar rice development has been highlighted in light navy blue color. ph phloem; c companion cell; nu nucellus; integ integuments (inner and outer); AC antipodal cells; CC central cell; SC synergid cell; EC egg cell; pn polar nuclei; vb vascular bundle; o ovule.

protocol. Each PCR cycle was conducted as follows: 94 °C for 10 s, 55 °C for 10 s, and 72 °C for 10 s, and relative quantification was calculated with *OsACT1* gene as a reference by the ΔΔCt method. Primer sequences used for PCR analyses are listed in Supplementary Table 1.

**Observation of rice grains by cryostat.** Between 7 and 10 days after flowering, rice grains of Nipponbare and the *osgcs1* mutant were fixed for ≥4 h in 4% paraformaldehyde, 5% acetic acid, and 50% ethanol. The grains were incubated in sequence with 10% (w/v) then 20% (w/v) sucrose in K-phosphate buffer (KPB) for 1 h each and 30% (w/v) sucrose in KPB overnight. The samples were embedded in Jung tissue freezing medium (Leica Microsystems Nussloch, Nussloch) and frozen at −80 °C. The grains were then bisected with a Leica CM1850 cryostat (Leica Biosystems Nussloch, Nussloch) at −20 °C and washed with distilled water. Thin sections were prepared with the cryostat (Leica Biosystems) at a thickness of 20 μm at −20 °C based on the Kawamoto method (Kawamoto, 2003). The sections were stained with 0.01% toluidine blue for 1 min and washed with distilled water. The samples were imaged with a stereomicroscope LW-820T (Wraymer, Osaka) and camera Wraycam CIX2000 (Wraymer, Osaka).

**Pollen tube observations for *osgcs1* mutants.** The method described by O'Donoughue and Bennett[28] was followed with modification. Pistil tissue, 24 h after pollination, was soaked in ethanol:acetic acid (3:1) and incubated for 30 min at room temperature. Fixed samples were soaked in lactic acid:ethanol (2:1) and autoclaved in Eppendorf tubes with the lids open at 120 °C for 20 min. Autoclaved samples were washed in 2% $K_3PO_4$ solution three times then soaked in 0.1% aniline blue in 2% $K_3PO_4$ solution and incubated for 2 h at room temperature.

**Transcriptome analysis.** Total RNA was treated with RNase-Free DNase I (Life Technologies) according to the manufacturer's instructions. The TruSeq RNA Sample Preparation Kit (Illumina) was used to construct complementary DNA (cDNA) libraries according to the manufacturer's instructions. The single ends of cDNA libraries were sequenced for 36 nucleotides from samples using the Illumina Genome Analyzer IIx (Illumina). The reads were mapped to the rice cDNA reference (Os-Nipponbare-Reference-IRGSP-1.0, https://rapdb.dna.affrc.go.jp/) using Bowtie[29] with the following options, "-v 3 -m 1 –all –best –strata," and the number of reads mapped to each reference was counted. The cluster tree illustrated in Supplementary Fig. 1 was based on hierarchical clustering analysis with Manhattan distance metrics and average linkage among samples. The correlation between two samples was derived as follows. First, genes without expression in any of the two samples were removed. Second, the logarithmic tag count of genes was used as input for the cor function of R (https://r-project.org/). The cluster was plotted by MeV (http://mev.tm4.org/). The loci which do not have gene ID were excluded from the following analysis. Early-response genes in the prefertilization event were collected by two categorizations, as shown in Fig. 2a: the value of WT_0DAP, a reference, was 0 or >0. For the latter case, the transcripts for which the value of WT_1DAP divided by the value of WT_0DAP was >3 were further screened. For each case, the transcripts that fulfilled all the following requirements were screened: the value of WT_3DAP divided by the value of WT_1DAP was <0.5; the value of *osgcs1*_3DAP divided by the value of *osgcs1*_1DAP was <0.5; and the value of *osgcs1*_1DAP divided by the value of WT_1DAP was >0.5. The genes associated with cell division, cyclins, cell expansion, expansins, and starch synthesis were manually identified and analyzed. The expression analysis was performed on genes related to starch and sucrose metabolism extracted from the KEGG PATHWAY database (https://www.genome.jp/kegg/pathway.html).

**Measurement of sugar accumulation by liquid chromatography-mass spectrometry.** The rice extracts were centrifuged at $15,000 \times g$ for 5 min and the supernatant was subjected to liquid chromatography-mass spectrometry (LC-MS) analysis (Dionex Ultimate 3000 HPLC system equipped with an autosampler, and an EXACTIVE Plus mass spectrometer [Thermo] fitted with an Unison UK-Amino column [3 μm, 4.6 × 250 mm; Imtakt]) at 60 °C using 1 mM aqueous ammonium acetate (pH 7.0) and acetonitrile (1:9, v/v) (500 μl min⁻¹, isocratic). Samples were detected in the ESI negative ion mode. Sugars were quantified using authentic standard sugars sucrose, glucose, and fructose as external standards. Data were acquired and analyzed with Xcalibur ver. 2.2 software [Thermo Electron Corporation].

**Statistics and reproducibility.** The *osgcs1* mutant plants used in the study were developed using CRISPR/Cas9 technology. Due to the hindered seed production in the mutants, they could not produce subsequent progeny generation. Hence, all of the analyses were carried out among the T0 lines. Each observational findings were confirmed by at least four replicates. The tissue-specific qRT-PCR analysis was carried out for two biological and six technical replications and analyzed by two-tailed *t*-test ($p \leq 0.001$). For the observation of watery seed-like structure, 6–10 panicles per mutant line were observed and the data were analyzed by Tukey-Kramer multiple comparison test ($p \leq 0.05, 0.005$). For the saccharides content and their proportions, four independent technical replicates with two biological samples (*osgcs1-g51* pollinated sugar-rice grains) each were carried out and the data were analyzed using Tukey's test ($p \leq 0.005$).

**Data availability**

All data needed to evaluate the conclusions in the paper are presented in the paper and/or the Supplementary Materials. Additional data related to this paper may be requested from the authors.

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

## Acknowledgements
We thank Etsuko Matsumoto, Naoko Iwata, Jiale He, and Chen Huang for technical assistance. We thank the National Institute of Genetics which provided the wild rice accessions and a Tos17 line with the support of the National Bioresource Project of the Ministry of Education, Culture, Sports, Science and Technology of Japan. We thank M. Ashikari and T. Higashiyama for critical discussion for this project. This work was supported by start-up funds from the School of Life Sciences, Fujian Agriculture and Forestry University (Grant #: 114-712018008 to R.D.K.) and the FAFU-UCR Joint Center, Haixia Institute of Science and Technology, Fujian Agriculture and Forestry University. This work was also supported by Chinese NSFC fund (Grant #: 31970809). This work was also supported by the Precursory Research for Embryonic Science and Technology (Grant #: 13416724 to R.D.K.; Kasahara Sakigake Project, Japan Science and Technology Agency). This work was partially supported by grants from Japan Society for the Promotion of Science Grants-in-Aid for Scientific Research (18KT0040) and the Cannon Foundation (R17-0070) to M.N.

## Author contributions
Y.H. and R.D.K. discovered the sugar-rice phenomenon. Y.H., K.B., and R.D.K. designed and created the sugar-rice and K.Kuwata analyzed the sugar contents in the sugar-rice. T.K. and K.Y. performed the phenotypic analysis for the sugar-rice by cryostat. Y.H., P.B.A., M.N., and K.Kurotani conducted the transcriptome analysis. Y.H., P.B.A., T.K., K.Y., X.L., S.Z., X.W., and R.D.K. performed phenotypic analysis for the rice POEM phenomenon. E.T. extracted mRNA from sperm cells from Nipponbare plants and performed qRT-PCR. Y.H. and R.D.K. designed the experiments. Y.H., P.B.A., M.N., and R.D.K., wrote the paper from the input of all authors' experimental results.

## Competing interests
The authors declare no competing interests.
