## [Peer Review File · Communications Biology]

Reviewers' comments:

Reviewer #1 (Remarks to the Author):

The manuscript, High-quality sugar production by fertilization-defective *osgcs1* rice authored by Honma et al., reports the creation of rice mutants that are defective in either one of two GCS1 genes and that produce enlarged ovules with high content of sugar. The transcriptome analysis of the mutant and wild type early ovules revealed down-regulation of genes involved in starch biosynthesis and sugar metabolism. One main concern from this reviewer is that the authors cited the some findings from their prior Arabidopsis work to presume that some aspects of the rice GCS1 genes work in the same way, which might be true or may not be true. For example, rice GCS1 mutants, unlike their Arabidopsis GCS1 mutant, have the unique enlarged ovules. Overall, the work does not provide much mechanistic basis of POEM except the transcriptome data.

Specific comments:

Please define GCS1 genes;

Extended Data Figure 1, The amino acids but not the nucleotides of genes were used for the phylogeny tree. Please change the names accordingly. Same with Extended Data Figure 2.

More information about and characterization of mutants for GCS1 and GCS1-Like are needed. It is very obvious that, based on the sequencing chromatograms (Sanger sequencing trace files) from g28, g83 and g51 for GCS1 and g41 for GCS1-like in Extended Data Figure 2a, the mutants were not homozygous mutants.

Lines 79-80: claiming both GCS1 and GCS1-like anther-specific is not conclusive as other tissues were not checked (Fig. 1b).

Lines 80-84, 86-90: Were sperm cells also released? Did fertilization occur? Can authors do reciprocal outcrossing of mutants and wild type? Genetic evidence for rice POEM is needed.

Lines 94-97: How the author explain the big variations between g28 and g83 and g51 if they are all knockouts?

Reviewer #2 (Remarks to the Author):

The comments to the paper can be found in the attached file.

Review Report

Title: “High-quality sugar production by fertilization-defective *osgcs1* rice”

Manuscript ID: COMMSBIO-20-0723-T

Journal: Nature Communications Biology

Dear Authors and Editor,

This paper claims the development of a new rice mutant line, which produces ovules containing 10% – 20% sugar, with a high level of sucrose content (98%). The authors suggest that these rice lines may be used as a sugar-producing crop line in alternative to sugarcane and sugar beet. The mutant lines produce these ovules as a consequence of failure in fertilization, although pollen tube contents are released inside the ovules. The authors suggest this to be the cause of ovule enlargement and sugar production, so ovules present POEM, a Pollen tube-dependent Ovule Enlargement Morphology. The POEM phenotype has been described before in more detail by Kasahara et al. 2016 (Science).

The novelty of this paper, besides the detection of POEMed ovules in a rice mutant line, relies mainly in the transcriptomic data analysis revealing some details on the accumulation of sugar by these ovules and the suggestion to use them as an alternative source of sugar, either for human consumption or for biofuel production. Overall, the experimental setup is well designed and structured, but there are some flaws in explaining the presented POEM phenotype. The authors should have analyzed the fertilization process, to concretely conclude about the POEMed ovules, strengthening the conclusions of this paper. Although the title of the paper is “High-quality sugar production by fertilization-defective *osgcs1* rice”, I do not consider that the fertilization-defective *osgcs1* rice phenotype has been fully proved, if considering only the presented results.

During the reading of this article I felt that the whole paper would greatly benefit from an English revision by a native speaking person. It is easy to read and understand but it presents some minor grammatical errors that need correction. This English review will surely benefit the MS making it clearer and much more pleasant to read.

The manuscript, indeed, shows significant findings relevant to the field and has the potential to be considered for publication at Communications Biology, but not in its present form. Therefore, I recommend major revisions.

Comments for the authors

1. Regarding the English comments, I show below some examples of what I think must be changed:

Lines 34 - 35 → "(...) which led fertilization failure (...)" substitute for "which led to fertilization failure"

Line 39 → "As transcriptome analysis revealed" substitute for "Trancryptomic analysis revealed that"

Line 40 → "had downregulation of starch biosynthesis genes" substitute for "had downregulation of starch biosynthetic genes"

Lines 40 - 41 → "have convert sucrose to" substitute for "have converted sucrose to"

Line 41 → "Overall study shows that" substitute for "Overall, this study shows that"

Line 43 → "to pave a way of developing novel" substitute for "to pave a way for developing novel"

Line 53 → "important source of" substitute for "important sources of"

Line 64 → It should be "Pollen tube content **release inside the ovule** triggers POEM (...)".

Lines 78 – 80 → "Since AtGCS1 is known to express in sperm cells, similar subcellular expression of OsGCS1 and OsGCS1-like genes is expected in rice as both of them showed anther-specificity." replace for "Since AtGCS1 is known to be expressed in the sperm cells, a similar subcellular expression of OsGCS1 and OsGCS1-like genes is expected in rice, as both of them have revealed to be anther-specific."

Lines 80 – 82 → "Pollinated rice style were stained with aniline blue, which showed that *osgcs1* pollen tubes were inserted correctly to the female gametophyte similar to Nipponbare pollen tubes, indicating that *osgcs1* mutants has no pollen tube guidance defect" must be changed for "Pollinated rice styles were stained with aniline blue, which showed that *osgcs1* pollen tubes have germinated and grown along the style until reaching the female gametophyte, in a similar way to what happen with the Nipponbare pollen tubes. These results indicate that *osgcs1* mutants have no pollen tube guidance defects (...)"

Lines 100 – 102 → “To distinguish the differences in gene regulatory network, we examined the gene expression profiles for Nipponbare and *osgcs1* (line g51) ovules using the Illumina sequencer (Fig. 2a to e).” please re-phrase it for a more self-explanatory sentence; start by explaining what you are comparing, how and why, e.g.: “The RNA-seq technology was used to analyze the transcriptomes of ovules from Nipponbare and *osgcs1* (line g51) plants, using the Illumina sequencer. These data was explored in order to better understand the differences in the gene regulatory networks involved in starch and sucrose metabolism (Fig. 2a to e).”

Line 111 → “Some of these genes showed a relatively lower expression (...)” Please replace “showed” for “revealed”, to avoid repetition of the word “showed”.

2. In Lines 78-80 the authors say that *OsGCS1* and *OsGCS1-like* have both revealed to be specifically expressed in the anthers, referring to Figure 1b. *OsGCS1* expression indeed seems to be anther-specific, but *OsGCS1-like* seems to be ovule specific, and not anther specific, as stated by the authors, although expressed in anthers in lower amounts. I suggest this information to be changed accordingly. Furthermore, only by performing an RT-PCR on rice sperm cells or by creating reporter lines for these two genes, *OsGCS1* and *OsGCS1-like*, the authors could prove if they are or not specifically expressed in the sperm cells. I think it is too imprudent to assume that these two genes are specific for the sperm cells just because they are expressed in the anthers (lines 78-80). If this is not possible, please change the text. It may be useful, for example, to refer that in the Japonica rice variety it has been shown the specific expression of *GCS1* (Os09g0525700) in sperm cells (Russell *et al.*, 2012, New Phytologist). And also to refer to the conservative nature of *GCS1* in reproduction HAP2(*GCS1*), and the appropriate bibliography.

3. Along the paper, and it is even written in the title and the summary, the authors keep stating that the ***gcs1* mutation in these rice lines led to a fertilization failure**. This has not been directly proved in any experiment shown in the MS. The paper will strongly benefit from a better depiction, in terms of experimental procedures, of the POEM phenotype. I feel this is poorly explored and described in this MS. Wouldn't it be possible to show the failure in fertilization by pollinating a rice line bearing female gametes fluorescently labeled such as the one from Ohnishi *et al.* 2014 (Plant Physiology) with *osgcs1* pollen? Or is this too difficult to perform and observe at the microscopic level?

The **aniline blue staining results** would definitely benefit from pictures showing the pollen tubes growing along the style; in that way the study could show us that there are no pollen tube guidance defects (line 82). The pictures shown in Figure 1 c and d should be presented as close-ups of pictures showing the ovule inside the pistil at a

lower magnification, where the pollen tubes may be observed growing along all pistil tissues. I suggest these images to be added to the MS. Moreover the WT (Nipponbare) image seems to show two pollen tubes entering the ovule micropyle, so, I would advise the authors to choose another image, if possible.

The observation of seeds and enlarged ovules (POEMed), reveal that these POEMed ovules have no embryo. But, the authors state in this paper that this lack of embryos in the *oscs1* rice ovules is caused by POEM, and consequently by the lack of fertilization. I am not completely convinced of this if the authors only show us these two aniline blue pictures. I know it is difficult to study reproduction in other plant species, the techniques may not be so easy to apply as in Arabidopsis plants, but detailed studies on fertilization have been performed in crop plants before, as for example in maize (Márton *et al.*, 2005, Science) and even in rice (Guo *et al.*, 2004, Protoplasma). So, I think that, showing that fertilization does not occur in these mutant lines would definitely strengthen the MS conclusions.

4. In the legend of Figure 1 please be clearer about the number of seeds observed for each line. It seems to be 10, but it is not clear. Please change it.

5. Lines 98-100. Please explain this better, the phrase is unclear. There are no results of Arabidopsis in this paper. If the authors refer to their previous study, please be clearer about this.

6. Line 106 “indicating that PTC release itself is sufficient to trigger biological events in rice ovules” – it would be more correct to say that “PTC release itself is sufficient to trigger transcriptional changes in rice ovules”.

7. Line 107 “As reported in Arabidopsis, multiple (...)” Please add reference.

8. Lines 109 – 110 “Since POEMed rice ovules showed no starch phenotype (Fig. 1g to h, and Extended Data fig. 3),” It will be helpful for the reader if this phenotype is introduced earlier when referring to this for the first time in the MS. For example, when describing the “watery seed-like” phenotype.

9. Line 129: “Figure 3 illustrates the proposed path of sugar rice is production.” Maybe replace for: “Figure 3 illustrates the proposed pathway by which sugar rice is produced.”

Extended text 1:

1. "Among various other genes, CYP78A13 was relatively highly up-regulated in Nipponbare ovules." Do you mean in *osgcs1* ovules?

2. "The gene has been reported to involve in checking embryo size and in facilitating endosperm size in rice." Please re-phrase this sentence to make it clearer.

Figure 1 f and h Legend: please refer the type of staining used.

Figure 3: I really like this Figure and I think it is perfect to illustrate the authors' hypothesis.

Response to Reviewer 1.

The manuscript, High-quality sugar production by fertilization-defective osgcs1 rice authored by Honma et al., reports the creation of rice mutants that are defective in either one of two GCSI genes and that produce enlarged ovules with high content of sugar. The transcriptome analysis of the mutant and wild type early ovules revealed down-regulation of genes involved in starch biosynthesis and sugar metabolism. One main concern from this reviewer is that the authors cited the some findings from their prior Arabidopsis work to presume that some aspects of the rice GCSI genes work in the same way, which might be true or may not be true. For example, rice GCSI mutants, unlike their Arabidopsis GCSI mutant, have the unique enlarged ovules. Overall, the work does not provide much mechanistic basis of POEM except the transcriptome data.

Dear Reviewer 1,

Thank you for your useful comments. During revision, we newly performed qRT-PCR using sperm cells along with additional tissues and added our previous data on the reciprocal crosses. As expected, we observed highly significant expression of both *OsGCSI* and *OsGCSI-like* in the sperm cells. According to your valuable suggestion, we have incorporated the results of reciprocal crosses, which show that the mutation is obtained only from the male side, in the revised manuscript. Additionally, we have changed the term “POEMed rice” to “POEMed-like rice” wherever appropriate after considering both reviewers’ suggestions. We have answered all of the questions hereunder. I sincerely appreciate your consideration.

Sincerely,

Ryushiro D. Kasahara

Specific comments:

Please define GCSI genes;

We defined the *GCSI* gene as follows,

“*GCSI* stands for generative cell specific 1 and refers to a gene that is expressed specifically in the sperm cells. It is required for double fertilization between sperm cells and the egg/central cells in *Arabidopsis*.”

Extended Data Figure 1, The amino acids but not the nucleotides of genes were used for the phylogeny tree. Please change the names accordingly. Same with Extended Data Figure 2.

We made changes as Reviewer 1 suggested.

More information about and characterization of mutants for GCSI and GCSI-Like are needed. It is very obvious that, based on the sequencing chromatograms (Sanger sequencing trace files) from g28, g83 and g51 for GCSI and g41 for GCSI-like in Extended Data Figure 2a, the mutants were not homozygous mutants.

Yes, the mutants were heterozygous. Since the sequences for Cas9 protein and gRNA were stably incorporated during transformation (see materials and method in extended text), we expected the other copy of the *GCSI* gene to be removed during plant growth or the fertilization process. However, as you stated, not all mutants gave the expected results. Based on the phenotypic data, we chose the mutant with the least variation and the best phenotypic characteristics (Fig. 1k).

We would like to answer the later question first because this question is strongly connected to this question.

Lines 94-97: How do the authors explain the big variations between g28 and g83 and g51 if they are all knockouts?

As mentioned earlier, the probable reason for this appears to be variation in the efficiency of the CRISPR-led excision of the remaining half-copy of the *GCSI* gene from pollens, which led to great variation among the mutants, as you stated. Since our study was focused on *GCSI* mutation-led seed phenotypic change (and not on CRISPR-led variations), we selected the mutant (*g51*) that showed the strongest phenotype and least variation (Fig. 1k) for further analytical observations.

Lines 79-80: claiming both GCSI and GCSI-like anther-specific is not conclusive as other tissues were not checked (Fig. 1b).

Thank you for pointing this out. We have recently obtained the qRT-PCR results for the respective RNA samples from rice sperm cells, ovules, and vegetative tissues (shoots as well as anthers) (Fig. 1b). The reviewer is correct in stating that *GCSI* and *GCSI-like* may not be anther-specific. As it happens, their expression levels in anthers (and shoots) were almost non-significant. However, their expression in the sperm cells was significantly higher than in the ovules (Fig. 1b). Therefore, we have rephrased the sentence to read “Since *AtGCSI* is known to be expressed in the sperm cells, a similar subcellular expression of *OsGCSI* and *OsGCSI-like* is expected in rice, as both of these genes show strong and highly significant expression in sperm cells.”

Lines 80-84, 86-90: Were sperm cells also released? Did fertilization occur? Can authors do reciprocal outcrossing of mutants and wild type? Genetic evidence for rice POEM is needed.

Since our study did not use plants with fluorescence-tagged sperm cells, we were not able to track it to confirm their release. However, by observing the pollen tube content (Fig. 1c and d), we strongly believe that release occurred based on our earlier experience with *Arabidopsis*. We observed that unlike WT ovules, *g51* ovules frequently attracted additional pollen tubes (a hallmark of failed fertilization by an earlier pollen tube) following pollen tube burst.

Additionally, we have added information on the reciprocal data to the main text as follows:

“We conducted the reciprocal crossing experiments using *osgcs1 (g51)* mutants and obtained 100% (n = 6 ovaries) fertilized seeds when the *osgcs1 (g51)* ovaries were crossed with Nipponbare pollen and 100% (n = 8 ovaries) seed-like tissue (as shown in Fig. 1g to j) when the Nipponbare ovaries were crossed with *osgcs1 (g51)* pollen, indicating that the mutation is transmitted through *osgcs1* male gametophytes.”

This result provides strong evidence that the mutation from the male side can produce a sugary rice grain.

Overall, we believe that the modifications made to the manuscript are sufficient, but not excessive.

Thank you very much for your valuable comments.

Response to Reviewer 2.

Dear Authors and Editor,

This paper claims the development of a new rice mutant line, which produces ovules containing 10%–20% sugar, with a high level of sucrose content (98%). The authors suggest that these rice lines may be used as a sugar-producing crop line in alternative to sugarcane and sugar beet. The mutant lines produce these ovules as a consequence of failure in fertilization, although pollen tube contents are released inside the ovules. The authors suggest this to be the cause of ovule enlargement and sugar production, so ovules present POEM, a Pollen tube-dependent Ovule Enlargement Morphology. The POEM phenotype has been described before in more detail by Kasahara et al. 2016 (Science).

*The novelty of this paper, besides the detection of POEMed ovules in a rice mutant line, relies mainly in the transcriptomic data analysis revealing some details on the accumulation of sugar by these ovules and the suggestion to use them as an alternative source of sugar, either for human consumption or for biofuel production. Overall, the experimental setup is well designed and structured, but there are some flaws in explaining the presented POEM phenotype. The authors should have analyzed the fertilization process, to concretely conclude about the POEMed ovules, strengthening the conclusions of this paper. Although the title of the paper is “High- quality sugar production by fertilization-defective *osgcs1* rice”, I do not consider that the fertilization-defective *osgcs1* rice phenotype has been fully proved, if considering only the presented results.*

During the reading of this article I felt that the whole paper would greatly benefit from an English revision by a native speaking person. It is easy to read and understand but it presents some minor grammatical errors that need correction. This English review will surely benefit the MS making it clearer and much more pleasant to read.

The manuscript, indeed, shows significant findings relevant to the field and has the potential to be considered for publication at Communications Biology, but not in its present form. Therefore, I recommend major revisions.

Dear Reviewer 2,

Thank you for the very supportive comments. Based on the suggestions, we recently performed the qRT-PCR experiments and obtained expression data for the *OsGCSI* and *OsGCSI-like* genes. Though not exclusively expressed in sperm cells as expected, we observed significantly high expression of these genes in sperm cells compared to ovules and all other tissues studied. We have added data from the reciprocal crossing experiments to the main text; these data further confirm that the male mutation was responsible for the phenotype. As mentioned below, we have excluded unnecessary emphasis on the relationships of POEMs or fertilization between *Arabidopsis* and rice so that the main finding of this study, i.e., a sugar-producing rice plant, would not be overshadowed.

We have answered all of the questions asked by Reviewer 2 below. I sincerely appreciate your consideration.

Sincerely,

Ryushiro D. Kasahara

Comments for the authors

1. Regarding the English comments, I show below some examples of what I think must be changed:

Lines 34 - 35 → “(...) which led fertilization failure (...)” substitute for “which led to fertilization failure”

Rephrased as Reviewer 2 suggested.

Line 39 → “As transcriptome analysis revealed” substitute for “Transcriptomic analysis revealed that”

Rephrased as Reviewer 2 suggested.

Line 40 → “had downregulation of starch biosynthesis genes” substitute for “had downregulation of starch biosynthetic genes”

Rephrased as Reviewer 2 suggested.

Lines 40 – 41 → “have convert sucrose to” substitute for “have converted sucrose to”

Rephrased as Reviewer 2 suggested.

Line 41 → “Overall study shows that” substitute for “Overall, this study shows that”

Rephrased as Reviewer 2 suggested.

Line 43 → “to pave a way of developing novel” substitute for “to pave a way for developing novel”

Rephrased as Reviewer 2 suggested.

Line 53 → “important source of” substitute for “important sources of”

Rephrased as Reviewer 2 suggested.

*Line 64 → It should be “Pollen tube content **release inside the ovule** triggers POEM*

(...)”.

Rephrased as Reviewer 2 suggested.

Lines 78 – 80 → “Since AtGCSI is known to express in sperm cells, similar subcellular expression of OsGCSI and OsGCSI-like genes is expected in rice as both of them showed anther-specificity.” replace for “Since AtGCSI is known to be expressed in the sperm cells, a similar subcellular expression of OsGCSI and OsGCSI-like genes is expected in rice, as both of them have revealed to be anther-specific.”

Recently, we conducted qRT-PCR experiments and found that the expression of both *OsGCSI* and *OsGCSI-like* was significantly higher in sperm cells than in ovules and all of the other tissues studied (revised Fig. 1b). Therefore, we rephrased the text to read “Since *AtGCSI* is known to be expressed at significantly high levels in the sperm cells, a similar subcellular expression of *OsGCSI* and *OsGCSI-like* is expected in rice, as both genes have been revealed to be sperm cell-specific.”

Lines 80 – 82 → “Pollinated rice style were stained with aniline blue, which showed that osgcs1 pollen tubes were inserted correctly to the female gametophyte similar to Nipponbare pollen tubes, indicating that osgcs1 mutants has no pollen tube guidance defect” must be changed for “Pollinated rice styles were stained with aniline blue, which showed that osgcs1 pollen tubes have germinated and grown along the style until reaching the female gametophyte, in a similar way to what happen with the Nipponbare pollen tubes. These results indicate that osgcs1 mutants have no pollen tube guidance defects (...)”

Rephrased as Reviewer 2 suggested.

Lines 100 – 102 → “To distinguish the differences in gene regulatory network, we examined the gene expression profiles for Nipponbare and osgcs1 (line g51) ovules using the Illumina

sequencer (Fig. 2a to e).” please re-phrase it for a more self-explanatory sentence; start by explaining what you are comparing, how and why, e.g.: “The RNA-seq technology was used to analyze the transcriptomes of ovules from Nipponbare and osgcs1 (line g51) plants, using the Illumina sequencer. These data was explored in order to better understand the differences in the gene regulatory networks involved in starch and sucrose metabolism (Fig. 2a to e).”

Rephrased as Reviewer 2 suggested.

Line 111 → “Some of these genes showed a relatively lower expression (...)” Please replace “showed” for “revealed”, to avoid repetition of the word “showed”.

Rephrased as Reviewer 2 suggested.

2. In Lines 78-80 the authors say that OsGCS1 and OsGCS1-like have both revealed to be specifically expressed in the anthers, referring to Figure 1b. OsGCS1 expression indeed seems to be anther-specific, but OsGCS1-like seems to be ovule specific, and not anther specific, as stated by the authors, although expressed in anthers in lower amounts. I suggest this information to be changed accordingly. Furthermore, only by performing an RT-PCR on rice sperm cells or by creating reporter lines for these two genes, OsGCS1 and OsGCS1-like, the authors could prove if they are or not specifically expressed in the sperm cells. I think it is too imprudent to assume that these two genes are specific for the sperm cells just because they are expressed in the anthers (lines 78-80). If this is not possible, please change the text. It may be useful, for example, to refer that sper in the Japonica rice variety it has been shown the specific expression of GCS1 (Os09g0525700) in m cells (Russell et al., 2012, New Phytologist). And also to refer to the conservative nature of GCS1 in reproduction HAP2(GCS1), and the appropriate bibliography.

Thank you for pointing this out. As per the reviewer’s suggestion, we carried out a fresh qRT-PCR analysis of the RNA samples separately derived from rice sperm cells, ovules, and representative vegetative tissues (stems and anthers). As the reviewer points out, we found

that the expression of both *OsGCSI* and *OsGCSI-like* was non-significant in anthers as well as in stems. However, the expression of these genes was significantly higher in sperm cells than in ovules (revised Fig. 1b). Therefore, we have rephrased the sentence to read “Since *AtGCSI* is known to be expressed in the sperm cells, a similar subcellular expression of *OsGCSI* and *OsGCSI-like* is expected in rice, as both of these genes show strong and highly significant expression in sperm cells.”

We also added the following sentence citing the article by Russell *et al.* as Reviewer 2 suggested strengthening the qRT-PCR result:

“A previous report¹⁸ showed the specific expression of *GCSI* (Os09g0525700) in the sperm cells of a Japonica rice variety, which completely matches our observations of the *OsGCSI* expression pattern.”

Additionally, we defined the *GCSI* gene as follows:

“*GCSI* stands for generative cell specific 1 and refers to a gene that is expressed specifically in the sperm cells. It is required for double fertilization between sperm cells and the egg/central cells in *Arabidopsis*.”

3. Along the paper, and it is even written in the title and the summary, the authors keep stating that the ***gcs1* mutation in these rice lines led to a fertilization failure**. This has not been directly proved in any experiment shown in the MS. The paper will strongly benefit from a better depiction, in terms of experimental procedures, of the POEM phenotype. I feel this is poorly explored and described in this MS. Wouldn't it be possible to show the failure in fertilization by pollinating a rice line bearing female gametes fluorescently labeled such as the one from Ohnishi *et al.* 2014 (*Plant Physiology*) with *oscgs1* pollen? Or is this too difficult to perform and observe at the microscopic level?

The ***aniline blue staining results*** would definitely benefit from pictures showing the pollen tubes growing along the style; in that way the study could show us that there are no pollen tube guidance defects (line 82). The pictures shown in Figure 1 c and d should be presented

as close-ups of pictures showing the ovule inside the pistil at a lower magnification, where the pollen tubes may be observed growing along all pistil tissues. I suggest these images to be added to the MS. Moreover the WT (Nipponbare) image seems to show two pollen tubes entering the ovule micropyle, so, I would advise the authors to choose another image, if possible.

*The observation of seeds and enlarged ovules (POEMed), reveal that these POEMed ovules have no embryo. But, the authors state in this paper that this lack of embryos in the *osgcs1* rice ovules is caused by POEM, and consequently by the lack of fertilization. I am not completely convinced of this if the authors only show us these two aniline blue pictures. I know it is difficult to study reproduction in other plant species, the techniques may not be so easy to apply as in *Arabidopsis* plants, but detailed studies on fertilization have been performed in crop plants before, as for example in maize (Márton et al., 2005, Science) and even in rice (Guo et al., 2004, Protoplasma). So, I think that, showing that fertilization does not occur in these mutant lines would definitely strengthen the MS conclusions.*

Thank you for pointing that out. We think that the reviewer is correct. Unfortunately, the studied specimen (*OsGCSI g51* mutant) is now expired since it only lasted for one generation due to a lack of fertility. Instead, we have toned down the wording throughout the whole manuscript so as not to emphasize that the mutants failed fertilization, and we have referred to the mutants as “POEMed-like rice.” We have even changed the title so as not to use the phrase “fertilization defective.”

Based on the homology search, qRT-PCR, and aniline blue experiments, we firmly believe that the *osgcs1* sperm cells failed fertilization, but we should not have used the term “fertilization defective” without direct evidence. We have changed “POEMed rice” to “POEM-like rice” throughout the manuscript. However, we may add the fertilization-related data as an addendum in the future if deemed necessary.

Additionally, we have added data on the *g51* mutant and Nipponbare to the main text as follows:

“We conducted the reciprocal crossing experiments using *osgcs1 (g51)* mutants and obtained 100% (n = 6 ovaries) fertilized seeds when the *osgcs1 (g51)* mutant was pollinated with Nipponbare pollen and 100% (n = 8 ovaries) seed-like tissue (as shown in Fig. 1g to j) when the Nipponbare ovaries were crossed with *osgcs1 (g51)* pollen, indicating that the mutation is transmitted from *osgcs1* male gametophytes. ”

This result provides strong evidence that the mutation from the male side can produce a sugary rice grain.

4. In the legend of Figure 1 please be clearer about the number of seeds observed for each line. It seems to be 10, but it is not clear. Please change it.

Rephrased as Reviewer 2 suggested as follows,

“(k) Watery seed-like tissue production due to *OsGCSI* or *OsGCSI-like* gene mutation. Ten seeds were observed for each mutant line. n = 10 (NB: Nipponbare), 10 (VA: vector alone), 5 (*g28*), 5 (*g83*), 6 (*g51*), 4 (Tos), and 5 (*g41*) lines, respectively.”

5. Lines 98-100. Please explain this better, the phrase is unclear. There are no results of *Arabidopsis* in this paper. If the authors refer to their previous study, please be clearer about this.

Rephrased as shown below,

“Since the *osgcs1* and *osgcs1-like* mutants showed the POEMed-like phenotype, as previously shown in POEMed *Arabidopsis*⁶, we hypothesized that there could be both shared and species-specific processes involved in the POEMed and/or POEMed-like phenomena in these two species. To test this hypothesis, RNA-seq technology was used to analyze the transcriptomes of ovules from Nipponbare and *osgcs1* (line *g51*) plants using an Illumina sequencer. The data acquired were further explored in order to better understand the

differences in the gene regulatory networks involved in starch and sucrose metabolism (Fig. 2a to e).”

We connected the phrase to the next two sentences.

6. Line 106 “*indicating that PTC release itself is sufficient to trigger biological events in rice ovules*” – *it would be more correct to say that “PTC release itself is sufficient to trigger transcriptional changes in rice ovules”.*

Rephrased as Reviewer 2 suggested.

7. Line 107 “*As reported in Arabidopsis, multiple (...)*” *Please add reference.*

Added the reference as Reviewer 2 suggested.

8. Lines 109 – 110 “*Since POEMed rice ovules showed no starch phenotype (Fig. 1g to h, and Extended Data fig. 3),*” *It will be helpful for the reader if this phenotype is introduced earlier when referring to this for the first time in the MS. For example, when describing the “watery seed-like” phenotype.*

Added the introduction as Reviewer 2 suggested as follows,

“Based on our observations, the watery seed-like tissue should contain no starch since the liquid was transparent. The contents of the tissue are discussed below...”

9. Line 129: “*Figure 3 illustrates the proposed path of sugar rice is production.*” *Maybe replace for: “Figure 3 illustrates the proposed pathway by which sugar rice is produced.”*

Rephrased as Reviewer 2 suggested.

Extended text 1:

1. *“Among various other genes, CYP78A13 was relatively highly up-regulated in Nipponbare ovules.” Do you mean in osgcs1 ovules?*

Thank you for pointing this out. Yes, it should read “*OsGCSI* ovules.” The necessary change has been made in the text.

2. *“The gene has been reported to involve in checking embryo size and in facilitating endosperm size in rice.” Please re-phrase this sentence to make it clearer.*

The sentence has been changed as follows:

“The gene reportedly plays a role in embryo size control and endosperm size facilitation in rice”

Figure 1 f and h Legend: *please refer the type of staining used.*

We have added the explanation as ‘(f, h) Toluidine blue staining.’

Figure 3: *I really like this Figure and I think it is perfect to illustrate the authors’ hypothesis.*

Thank you for your compliments. We hoped to summarize the whole concept in this figure as well. Based on your comment, we believe that our efforts were worthwhile.

Overall, we hope that the changes we have made to the manuscript are appropriate and not too excessive for your review. We have followed the reviewer's recommendations in all cases. We certainly appreciate your creative suggestions.

Thank you very much.

REVIEWERS' COMMENTS:

Reviewer #1 (Remarks to the Author):

The authors have addressed this reviewer's previous comments satisfactorily and the revised manuscript has been significantly improved.

Reviewer #2 (Remarks to the Author):

Dear authors and Editor,

Overall, and taking into account the comments I have made in the first revision, I feel that the authors have addressed all the suggestions and comments appropriately. The main focus of the manuscript changed from the fertilization defective phenotype to the production of sugary rice ovules, which I think is wiser, benefiting the manuscript. The reciprocal crosses experiment, without any doubt, gave more strength to the described results.

The manuscript reveals important findings to the field and has the potential to be considered for publication at Communications Biology.